# MULTI-TIMESCALE REPRESENTATION LEARNING IN LSTM LANGUAGE MODELS

**Shivangi Mahto**[*]
Department of Computer Science
The University of Texas at Austin
Austin, TX, USA
shivangi@utexas.edu

**Vy A. Vo**
Brain-Inspired Computing Lab
Intel Labs
Hillsboro, OR, USA
vy.vo@intel.com

**Javier S. Turek**
Brain-Inspired Computing Lab
Intel Labs
Hillsboro, OR, USA
javier.turek@intel.com

**Alexander G. Huth**
Depts. of Computer Science & Neuroscience
The University of Texas at Austin
Austin, TX, USA
huth@cs.utexas.edu

## ABSTRACT

Language models must capture statistical dependencies between words at timescales ranging from very short to very long. Earlier work has demonstrated that dependencies in natural language tend to decay with distance between words according to a power law. However, it is unclear how this knowledge can be used for analyzing or designing neural network language models. In this work, we derived a theory for how the memory gating mechanism in long short-term memory (LSTM) language models can capture power law decay. We found that unit timescales within an LSTM, which are determined by the forget gate bias, should follow an Inverse Gamma distribution. Experiments then showed that LSTM language models trained on natural English text learn to approximate this theoretical distribution. Further, we found that explicitly imposing the theoretical distribution upon the model during training yielded better language model perplexity overall, with particular improvements for predicting low-frequency (rare) words. Moreover, the explicit multi-timescale model selectively routes information about different types of words through units with different timescales, potentially improving model interpretability. These results demonstrate the importance of careful, theoretically-motivated analysis of memory and timescale in language models.

## 1 INTRODUCTION

Autoregressive language models are functions that estimate a probability distribution over the next word in a sequence from past words, $p(w_t|w_{t-1}, \ldots, w_1)$. This requires capturing statistical dependencies between words over short timescales, where syntactic information likely dominates (Adi et al., 2017; Linzen et al., 2016), as well as long timescales, where semantic and narrative information likely dominate (Zhu et al., 2018; Conneau et al., 2018; Gulordava et al., 2018). Because this probability distribution grows exponentially with sequence length, some approaches simplify the problem by ignoring long-range dependencies. Classical $n$-gram models, for example, assume word $w_t$ is independent of all but the last $n-1$ words, with typical $n=5$ (Heafield, 2011). Hidden Markov models (HMMs) assume that the influence of previous words decays exponentially with distance from the current word (Lin & Tegmark, 2016).

In contrast, neural network language models such as recurrent (Hochreiter & Schmidhuber, 1997; Merity et al., 2018; Melis et al., 2018) and transformer networks (Melis et al., 2019; Krause et al., 2019; Dai et al., 2019) include longer-range interactions, but simplify the problem by working in lower-dimensional representational spaces. Attention-based networks combine position and

---

[*]Current affiliation: Apple Inc.

content-based information in a small number of attention heads to flexibly capture different types of dependencies within a sequence (Vaswani et al., 2017; Cordonnier et al., 2019). Gated recurrent neural networks (RNNs) compress information about past words into a fixed-length state vector (Hochreiter & Schmidhuber, 1997). The influence each word has on this state vector tends to decay exponentially over time. However, each element of the state vector can have a different exponential time constant, or "timescale" (Tallec & Ollivier, 2018), enabling gated RNNs like the long short-term memory (LSTM) network to flexibly learn many different types of temporal relationships (Hochreiter & Schmidhuber, 1997). Stacked LSTM networks reduce to a single layer (Turek et al., 2020), showing that network depth has an insignificant influence on how the LSTM captures temporal relationships. Yet in all these networks the shape of the temporal dependencies must be learned directly from the data. This seems particularly problematic for very long-range dependencies, which are only sparsely informative (Lin & Tegmark, 2016).

This raises two related questions: what should the temporal dependencies in a language model look like? And how can that information be incorporated into a neural network language model?

To answer the first question, we look to empirical and theoretical work that has explored the dependency statistics of natural language. Lin & Tegmark (2016) quantified temporal dependencies in English and French language corpora by measuring the mutual information between tokens as a function of the distance between them. They observed that mutual information decays as a power law, i.e. $MI(w_k, w_{k+t}) \propto t^{-d}$ for constant $d$. This behavior is common to hierarchically structured natural languages (Lin & Tegmark, 2016; Sainburg et al., 2019) as well as sequences generated from probabilistic context-free grammars (PCFGs) (Lin & Tegmark, 2016).

Now to the second question: if temporal dependencies in natural language follow a power law, how can this information be incorporated into neural network language models? To our knowledge, little work has explored how to control the temporal dependencies learned in attention-based models. However, many approaches have been proposed for controlling gated RNNs, including updating different groups of units at different intervals (El Hihi & Bengio, 1996; Koutnik et al., 2014; Liu et al., 2015; Chung et al., 2017), gating units across layers (Chung et al., 2015), and explicitly controlling the input and forget gates that determine how information is stored and removed from memory (Xu et al., 2016; Shen et al., 2018; Tallec & Ollivier, 2018). Yet none of these proposals incorporate a specific shape of temporal dependencies based on the known statistics of natural language.

In this work, we build on the framework of Tallec & Ollivier (2018) to develop a theory for how the memory mechanism in LSTM language models can capture temporal dependencies that follow a power law. This relies on defining the *timescale* of an individual LSTM unit based on how the unit retains and forgets information. We show that this theory predicts the distribution of unit timescales for LSTM models trained on both natural English (Merity et al., 2018) and formal languages (Suzgun et al., 2019). Further, we show that forcing models to follow this theoretical distribution improves language modeling performance. These results highlight the importance of combining theoretical modeling with an understanding of how language models capture temporal dependencies over multiple scales.

## 2 MULTI-TIMESCALE LANGUAGE MODELS

### 2.1 TIMESCALE OF INFORMATION

We are interested in understanding how LSTM language models capture dependencies across time. Tallec & Ollivier (2018) elegantly argued that memory in individual LSTM units tends to decay exponentially with a time constant determined by weights within the network. We refer to the time constant of that exponential decay as the unit's representational timescale.

Timescale is directly related to the LSTM memory mechanism (Hochreiter & Schmidhuber, 1997), which involves the LSTM cell state $c_t$, input gate $i_t$ and forget gate $f_t$,

$$
\begin{aligned}
i_t &= \sigma(W_{ix}x_t + W_{ih}h_{t-1} + b_i) \\
f_t &= \sigma(W_{fx}x_t + W_{fh}h_{t-1} + b_f) \\
\tilde{c}_t &= \tanh(W_{cx}x_t + W_{ch}h_{t-1} + b_c) \\
c_t &= f_t \odot c_{t-1} + i_t \odot \tilde{c}_t,
\end{aligned}
$$

where $x_t$ is the input at time $t$, $h_{t-1}$ is the hidden state, $W_{ih}, W_{ix}, W_{fh}, W_{fx}, W_{ch}, W_{cx}$ are the different weights and $b_i, b_f, b_c$ the respective biases. $\sigma(\cdot)$ and $\tanh(\cdot)$ represent element-wise sigmoid and hyperbolic tangent functions. Input and forget gates control the flow of information in and out of memory. The forget gate $f_t$ controls how much memory from the last time step $c_{t-1}$ is carried forward to the current state $c_t$. The input gate $i_t$ controls how much information from the input $x_t$ and hidden state $h_{t-1}$ at the current timestep is stored in memory for subsequent timesteps.

To find the representational timescale, we consider a "free input" regime with zero input to the LSTM after timestep $t_0$, i.e., $x_t = 0$ for $t > t_0$. Ignoring information leakage through the hidden state (i.e., assuming $W_{ch} = 0$, $b_c = 0$, and $W_{fh} = 0$) the cell state update becomes $c_t = f_t \odot c_{t-1}$. For $t > t_0$, it can be further simplified as

$$
\begin{aligned}
c_t &= f_0^{t-t_0} \odot c_0 \\
&= e^{(\log f_0)(t-t_0)} \odot c_0,
\end{aligned}
\tag{1}
$$

where $c_0 = c_{t_0}$ is the cell state at $t_0$, and $f_0 = \sigma(b_f)$ is the value of the forget gate, which depends only on the forget gate bias $b_f$. Equation 1 shows that LSTM memory exhibits exponential decay with characteristic *forgetting time*

$$
T = -\frac{1}{\log f_0} = \frac{1}{\log(1 + e^{-b_f})}.
\tag{2}
$$

That is, values in the cell state tend to shrink by a factor of $e$ every $T$ timesteps. We refer to the forgetting time in Equation 2 as the representational timescale of an LSTM unit.

Beyond the "free input" regime, we can estimate the timescale for a LSTM unit by measuring the average forget gate value over a set of test sequences,

$$
T_{est} = -\frac{1}{\log \bar{f}},
\tag{3}
$$

where $\bar{f} = \frac{1}{KN} \sum_{j=1}^{N} \sum_{t=1}^{K} f_t^j$, in which $f_t^j$ is the forget gate value of the unit at $t$-th timestep for $j$-th test sequence, $N$ is the number of test sequences, and $K$ is the test sequence length.

## 2.2 COMBINING EXPONENTIAL TIMESCALES TO YIELD A POWER LAW

From earlier work, we know that temporal dependencies in natural language tend to decay following a power law (Lin & Tegmark, 2016; Sainburg et al., 2019). Yet from Equation 1 we see that LSTM memory tends to decay exponentially. These two decay regimes are fundamentally different–the ratio of a power law divided by an exponential always tends towards infinity. However, LSTM language models contain many units, each of which can have a different timescale. Thus LSTM language models might approximate power law decay through a combination of exponential functions. Here we derive a theory for how timescales should be distributed within an LSTM in order to yield overall power law decay.

Let us assume that the timescale $T$ for each LSTM unit is drawn from a distribution $P(T)$. We want to find a $P(T)$ such that the expected value over $T$ of the function $e^{\frac{-t}{T}}$ approximates a power law decay $t^{-d}$ for some constant $d$,

$$
t^{-d} \propto \mathbb{E}_T[e^{-t/T}] = \int_0^\infty P(T) e^{-t/T} dT.
\tag{4}
$$

Solving this problem reveals that $P(T)$ is an *Inverse Gamma distribution* with shape parameter $\alpha = d$ and scale parameter $\beta = 1$ (see Section A.1 for derivation). The probability density function of the Inverse Gamma distribution is given as $P(T; \alpha, \beta) = \frac{\beta^\alpha}{\Gamma(\alpha)} (1/T)^{\alpha+1} e^{(-\beta/T)}$.

This theoretical result suggests that in order to approximate the power law decay of information in natural language, unit timescales in LSTM language models should follow an Inverse Gamma distribution. We next perform experiments to test whether this prediction holds true for models trained on natural language and models trained on samples from a formal language with known temporal statistics. We then test whether enforcing an Inverse Gamma timescale distribution at training time improves model performance.

## 2.3 Controlling LSTM unit timescales

To enforce a specific distribution of timescales in an LSTM and thus create an explicit multi-timescale model, we drew again upon the methods developed in Tallec & Ollivier (2018). Following the analysis in Section 2.1, the desired timescale $T_{desired}$ for an LSTM unit can be controlled by setting the forget gate bias to the value

$$b_f = -\log(e^{\frac{1}{T_{desired}}} - 1). \tag{5}$$

The balance between forgetting information from the previous timestep and adding new information from the current timestep is controlled by the relationship between forget and input gates. To maintain this balance we set the input gate bias $b_i$ to the opposite value of the forget gate, i.e., $b_i = -b_f$. This ensures that the relation $i_t \approx 1 - f_t$ holds true. Importantly, these bias values remain fixed (i.e. are *not* learned) during training, in order to keep the desired timescale distribution across the network.

## 3 Evaluation

### 3.1 Experimental Setup

Here we describe our experimental setup. We examined two regimes: natural language data, and synthetic data from a formal language. All models were implemented in pytorch (Paszke et al., 2019) and the code can be downloaded from `https://github.com/HuthLab/multi-timescale-LSTM-LMs`.

#### 3.1.1 Natural Language

We experimentally evaluated LSTM language models trained on the Penn Treebank (PTB) (Marcus et al., 1999; Mikolov et al., 2011) and WikiText-2 (WT2) (Merity et al., 2017) datasets. PTB contains a vocabulary of 10K unique words, with 930K tokens in the training, 200K in validation, and 82K in test data. WT2 is a larger dataset with a vocabulary size of 33K unique words, almost 2M tokens in the training set, 220K in the validation set, and 240K in the test set. As a control, we also generated a Markovian version of the PTB dataset. Using the empirical bigram probabilities from PTB, we sampled tokens sequentially until a new corpus of the same size had been generated.

We compared two language models: a standard stateful LSTM language model (Merity et al., 2018) as the baseline, and our multi-timescale language model. Both models comprise three LSTM layers with 1150 units in the first two layers and 400 units in the third layer, with an embedding size of 400. Input and output embeddings were tied. All models were trained using SGD followed by non-monotonically triggered ASGD for 1000 epochs. Training sequences were of length 70 with a probability of 0.95 and 35 with a probability of 0.05. During inference, all test sequences were length 70. For training, all embedding weights were uniformly initialized in the interval $[-0.1, 0.1]$. All weights and biases of the LSTM layers in the baseline language model were uniformly initialized between $\left[\frac{-1}{H}, \frac{1}{H}\right]$ where $H$ is the output size of the respective layer.

The multi-timescale language model has the same initialization, except for the forget and input gate bias values that are assigned following Equation 5 and fixed during training. For layer 1, we assigned timescale $T = 3$ to half the units and $T = 4$ to the other half. For layer 2, we assigned timescales to each unit by selecting values from an Inverse Gamma distribution. The shape parameter of the distribution was set to $\alpha = 0.56$ after testing different values (see A.5). The timescales in this layer had $80\%$ of the units below 20, and the rest ranging up to the thousands. For layer 3, the biases were initialized like the baseline language model and trained (not fixed).

#### 3.1.2 Formal Language: the Dyck-2 Grammar

We also tested the LSTM language models on a formal language with known temporal statistics. The Dyck-2 grammar (Suzgun et al., 2019) defines all possible valid sequences using two types of parentheses. Because Dyck-2 is a probabilistic context-free grammar (PCFG), its temporal dependencies will tend to decay following a power law (Lin & Tegmark, 2016). Further, timescales within particular sequences can be measured as the distance between matching opening and closing parenthesis pairs. We randomly sampled from the grammar with probabilities $p_1 = 0.25$, $p_2 = 0.25$,

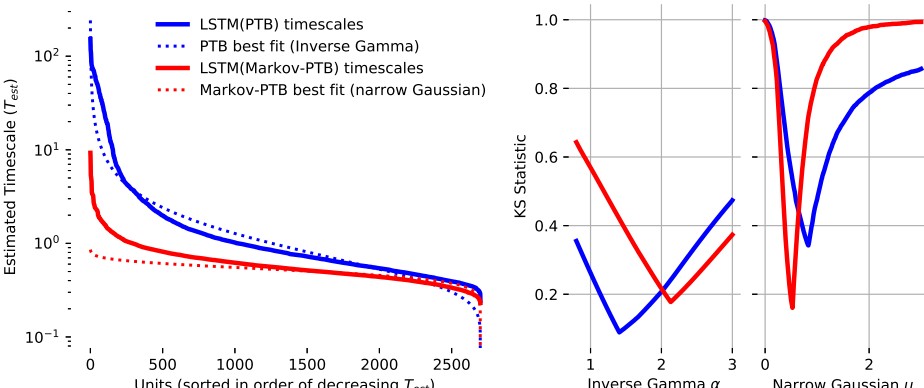

Figure 1: Empirical timescale distributions and best distribution fits for an LSTM language model trained on natural language (PTB; blue), and the same model trained on a Markovian corpus (Markov-PTB; red). Left: An Inverse Gamma distribution is the best fit for the PTB corpus (dotted blue), while a narrow Gaussian is the best fit for the Markovian corpus (dotted red). Right: We determined the best fits by varying distribution parameters ($\alpha$ for the Inverse Gamma, $\mu$ for the narrow Gaussian) and computing the Kolmogorov-Smirnov statistic to measure the difference between the empirical and theoretical distributions (lower is better). The LSTM trained on natural language (blue) is best fit by the Inverse Gamma distribution with $\alpha = 1.4$, while the LSTM trained on a Markovian corpus (red) is best fit by the narrow Gaussian distribution with $\mu = 0.5$.

and $q = 0.25$ to generate training, validation, and test sets with 10K, 2K, and 5K sequences. Sequences contain a maximum of 200 elements.

To solve this task, we followed Suzgun et al. (2019). Models were trained to predict which closing parentheses could appear next. Elements in the sequence were one-hot encoded and fed as input to the models. We trained a baseline model and a multi-timescale version. Each model consists of a 256-unit LSTM layer followed by a linear output layer. The output layer reduces the dimension to 2, the number of possible closing parentheses. These output values are each converted to a $(0, 1)$ interval with a sigmoid function. Parameters were initialized using a uniform distribution. The multi-timescale model has the forget and input gate bias values assigned using an Inverse Gamma distribution with $\alpha = 1.50$, and fixed during training. This parameter best matched the measured distribution of timescales. Training minimized the mean squared error (MSE) loss using the Adam optimizer (Kingma & Ba, 2015) with learning rate $1e-4$, $\beta_1 = 0.9$, $\beta_2 = 0.999$, and $\varepsilon = 1e-8$ for 2000 epochs.

## 3.2 EXPERIMENTAL RESULTS

### 3.2.1 DO EMPIRICAL LSTM UNIT TIMESCALES APPROXIMATE LANGUAGE STATISTICS?

In Section 2.2 we predicted that the distribution of unit timescales within an LSTM language model should follow an Inverse Gamma distribution. To test this prediction, we trained an LSTM language model on a natural language corpus (PTB), and then trained the same model on a Markovian dataset generated with the bigram statistics of PTB (Markov-PTB). We estimated the timescale of each unit using Equation 3. While representing PTB should require an Inverse Gamma distribution of timescales, representing Markov-PTB should only require a single timescale (i.e. a delta function, which can be approximated as a narrow Gaussian with $\sigma$=0.1). We tested this by generating a family of Inverse Gamma distributions or narrow Gaussian functions, and measuring the difference between the empirical distributions and these theoretical distributions. This was quantified with the Kolmogorov-Smirnov test statistic (Massey Jr, 1951).

The results in Figure 1 suggest that our theoretical prediction is correct. The Inverse Gamma distribution better fits the model trained on natural language, while the approximate delta function (narrow Gaussian distribution) better fits the model trained on the Markovian corpus.

Table 1: Perplexity of the multi-timescale and baseline models for tokens across different frequency bins for the Penn TreeBank (PTB) and WikiText-2 (WT2) test datasets. We also report the mean difference in perplexity (baseline − multi-timescale) across 10,000 bootstrapped samples, along with the 95% confidence interval (CI).

**Dataset:** Penn TreeBank

| Model | above 10K | 1K-10K | 100-1K | below 100 | All tokens |
|---|---|---|---|---|---|
| Baseline | 6.82 | 27.77 | 184.19 | 2252.50 | 61.40 |
| Multi-timescale | 6.84 | 27.14 | 176.11 | 2100.89 | 59.69 |
| Mean diff. | -0.02 | 0.63 | 8.08 | 152.03 | 1.71 |
| 95% CI | [-0.06, 0.02] | **[0.38, 0.88]** | **[6.04, 10.2]** | **[119.1,186.0]** | **[1.41, 2.02]** |

**Dataset:** WikiText-2

| Model | above 10K | 1K-10K | 100-1K | below 100 | All tokens |
|---|---|---|---|---|---|
| Baseline | 7.49 | 49.70 | 320.59 | 4631.08 | 69.88 |
| Multi-timescale | 7.46 | 48.52 | 308.43 | 4318.72 | 68.08 |
| Mean diff. | 0.03 | 1.17 | 12.20 | 312.13 | 1.81 |
| 95% CI | **[0.01,0.06]** | **[0.83,1.49]** | **[9.96,14.4]** | **[267.9,356.3]** | **[1.61,2.01]** |

### 3.2.2 DOES IMPOSING AN INVERSE GAMMA DISTRIBUTION IMPROVE MODEL PERFORMANCE?

The previous section showed that the empirical distribution of timescales in an LSTM LM trained on natural language is well-approximated by an Inverse Gamma distribution, as predicted. However, the model had to learn this distribution from a small and noisy dataset. If the Inverse Gamma distribution is indeed optimal for capturing power law dependencies, then it should be beneficial to simply enforce this distribution. To test this hypothesis, we constructed models with enforced Inverse Gamma timescales, which we call multi-timescale (MTS) language models (see Section 3.1.1 for details). Compared to the baseline LSTM model, the multi-timescale model incorporates a timescale prior that matches the statistical temporal dependencies of natural language. The multi-timescale model is particularly enriched in long timescale units (see A.5).

If this is an effective and useful prior, the multi-timescale model should perform better than the baseline when very long timescales are needed to accomplish a task. We first compared total language modeling performance between the baseline and multi-timescale models. The overall perplexities on the test datasets are shown in the far right column of Table 1. The multi-timescale language model outperforms the baseline model for both datasets by an average margin of 1.60 perplexity. Bootstrapping over the test set showed that this improvement is statistically significant. Test data were divided into 100-word sequences and resampled with replacement 10,000 times. For each sample, we computed the difference in model perplexity (baseline − multi-timescale) and reported the 95% confidence intervals (CI) in Table 1. Differences are significant at $p < 0.05$ if the CI does not overlap with 0.

Previous work has demonstrated that common, typically closed-class, words rely mostly on short timescale information, whereas rare, typically open-class, words require longer timescale information (Khandelwal et al., 2018; Griffiths et al., 2005; Rosenfeld, 1994; Iyer & Ostendorf, 1996). To test whether the improved performance of the multi-timescale model could be attributed to the very long timescale units, we computed model perplexities across different word frequency bins. We hypothesized that the multi-timescale model should show the greatest improvements for infrequent words, while showing little to no improvement for common words.

We divided the words in the test dataset into 4 bins depending on their frequencies in the training corpus: more than 10,000 occurrences; 1000-10,000 occurrences; 100-1000 occurrences; and fewer than 100 occurrences. Then we compared performance of the models for words in each bin in Table 1. The multi-timescale model performed significantly better than baseline in both datasets for the 2 less frequent bins (bootstrap test), with increasing difference for less frequent words. This suggests that the performance advantage of the multi-timescale model is highest for infrequent words, which require very long timescale information.

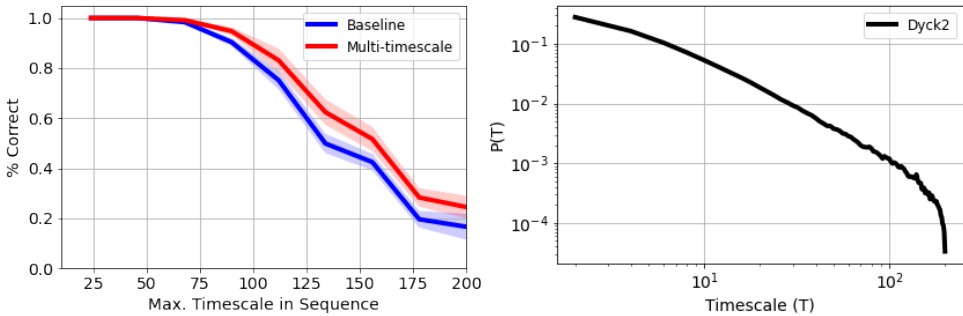

Figure 2: The left plot shows the performance of the baseline (blue) and multi-timescale (red) models on the Dyck-2 grammar as a function of the maximum timescale in a sequence (20 repetitions, standard deviation). A correctly predicted sequence is considered to have all the outputs correct. The graph shows that the multi-timescale model is predicting better up to 10% more sequences in the longer timescales regime ($T > 75$). The right plot describe the distribution of all timescales as computed from the training data. The timescale distribution decays as a power law. The sharp decay near $T = 200$ is due to the limitation of the sequence lengths imposed to generate the training data.

These results strongly support our claim that the Inverse Gamma timescale distribution is a good prior for models trained on natural language that is known to have power law temporal dependencies.

### 3.2.3 GENERALIZATION TO FORMAL LANGUAGES

Complex dependencies in natural language data may not necessarily follow any precise statistical law, so we also turned to formal languages to evaluate our hypotheses. In this experiment we used the Dyck-2 grammar, which enables us to precisely measure the distribution of timescales within the data (see Figure 2 right). We first used this capability to find the best $\alpha$ parameter value for the Inverse Gamma distribution fitting a power law function, and then to measure model performance as a function of timescale.

Following Suzgun et al. (2019), baseline and multi-timescale language models were trained to predict closing parenthesis types. A sequence is considered correctly predicted when every output from the model is correct across the entire sequence. The baseline model was able to correctly predict the output of 91.66% of test sequences, while the multi-timescale model achieved 93.82% correct. Figure 2 (left) shows percent correct as a function of the maximum timescale in a sequence over 20 training repetitions. Both models succeeded at predicting most sequences shorter than 75 steps. However, for longer timescales, the multi-timescale model correctly predicted 5-10% more sequences than the baseline model. This result supplements the idea that enforcing Inverse Gamma timescales improves the model's ability to capture longer-range dependencies. Nevertheless, both models are far from perfect on longer timescales, raising the question of how to further improve these longer timescales.

### 3.2.4 ROUTING OF INFORMATION THROUGH LSTM UNITS WITH DIFFERENT TIMESCALES

Our previous results demonstrated successful control the timescale distribution in our multi-timescale model, and that this improved model performance. Breaking out performance results by word frequency also showed that the multi-timescale model was better able to capture some words–specifically, low-frequency words–than the baseline LSTM. This suggests that the multi-timescale model, which is enriched in long-timescale units relative to the baseline model, may be selectively routing information about infrequent words through specific long-timescale units. More broadly, the multi-timescale model could be selectively routing information to each unit based on its timescale. This could make the multi-timescale model more interpretable, since at least one property of each unit–its timescale–would be known ahead of time. To test this hypothesis, we again divided the test data into word frequency bins. If long timescale information is particularly important for low frequency words, then we would expect information about those words to be selectively routed through long timescale units. We tested the importance of each LSTM unit for words in each

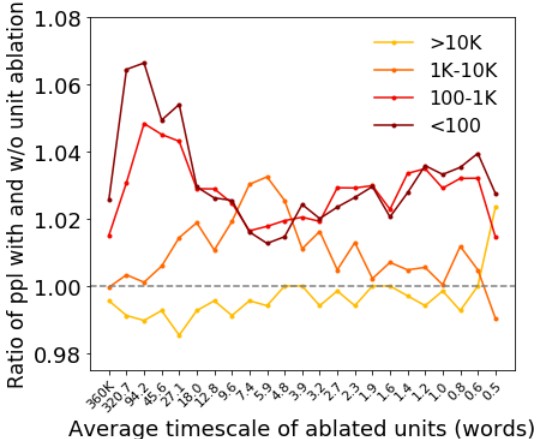

Figure 3: Information routing across different units of the multi-timescale LSTM for PTB dataset. Each line shows results for words in a different frequency bin (i.e. rare words occur $< 100$ times, red line). The ordinate axis shows the ratio of model perplexity with and without ablation of a group of 50 LSTM units, sorted and grouped by assigned timescale. Ratios above 1 indicate a decrease in performance following ablation, suggesting that the ablated units are important for predicting words in the given frequency bin. Abscissa shows the average timescale of each group (left is longer timescale).

frequency bin by selectively ablating the units during inference, and then measuring the effect on prediction performance.

We divided the LSTM units from layer 2 of the multi-timescale model into 23 groups of 50 consecutive units, sorted by assigned timescale. We ablated one group of units at a time by explicitly setting their output to 0, while keeping the rest of the units active. We then computed the model perplexity for different word frequency bins, and plotted the ratio of the perplexity with and without ablation. If performance gets worse for a particular word frequency bin when ablating a particular group of units, it implies that the ablated units are important for predicting those words.

Figure 3 shows this ratio across all frequency bins and groups for the PTB dataset (similar results for WikiText-2 are shown in the supplement). Ablating units with long timescales (integrating over 20-300 timesteps) causes performance to degrade the most for low frequency words (below 100 and 1K-10K occurences); ablating units with medium timescales (5-10 timesteps) worsens performance for medium frequency words (1K-10K occurrences); and ablating units with the shortest timescales ($<$1 timestep) resulted in worse performance on the highest frequency words. These results demonstrate that timescale-dependent information is routed through different units in this model, suggesting that the representations that are learned for different timescales are interpretable.

## 4 CONCLUSION

In this paper we developed a theory for how LSTM language models can capture power law temporal dependencies. We showed that this theory predicts the distribution of timescales in LSTM language models trained on both natural and formal languages. We also found that explicit multi-timescale models that are forced to follow this theoretical distribution give better performance, particularly over very long timescales. Finally, we show evidence that information dependent on different timescales is routed through specific units, demonstrating that the unit timescales are highly interpretable. This enhanced interpretability makes it possible to use LSTM activations to predict brain data, as in (Jain & Huth, 2018), and estimate processing timescales for different brain regions (Jain et al., 2020). These results highlight the importance of theoretical modeling and understanding of how language models capture dependencies over multiple timescales.

ACKNOWLEDGMENTS

We would like to thank Shailee Jain for valuable feedback on the manuscript and useful discussions, and the anonymous reviewers for their insights and suggestions. Funding support for this work came from the Burroughs Wellcome Fund Career Award at the Scientific Interface (CASI), Intel Research Award, and Alfred P. Sloan Foundation Research Fellowship.

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

# A  SUPPLEMENTARY MATERIAL

## A.1  DERIVATION OF INVERSE GAMMA TIMESCALE DISTRIBUTION

In Section 2.2 we examined how exponentials can be additively combined to yield a power law. Here we give a detailed derivation of how choosing exponential timescales according to the Inverse Gamma distribution with shape parameter $\alpha = d$ and scale parameter $\beta = 1$ yields power law decay with exponent $-d$.

Recall that our goal is to identify the probability distribution over exponential timescales $P(T)$ such that the expected value over $T$ of the exponential decay function $e^{-\frac{t}{T}}$ approximates a power law decay $t^{-d}$ with some constant $d$,

$$t^{-d} \propto \mathbb{E}_T[e^{-t/T}] = \int_0^\infty P(T)e^{-\frac{t}{T}}dT.$$

Noting its similarity to the integral we are attempting to solve, we begin with the definition of the Gamma function $\Gamma(a)$,

$$\Gamma(a) = \int_0^\infty u^{a-1}e^{-u}du. \tag{6}$$

Now we perform the substitution $u(T) = t/T$ inside the integral, assuming that $t \geq 0$ and $T > 0$. This substitution makes $du = \frac{-t}{T^2}dT$, and also changes the limits of integration from $(0, \infty)$ to $(u(0), u(\infty)) = (\infty, 0)$, giving

$$\Gamma(a) = \int_\infty^0 \left(\frac{t}{T}\right)^{a-1} e^{-\frac{t}{T}} \left(\frac{-t}{T^2}dT\right) \tag{7}$$

$$= -\int_\infty^0 \left(t^{a-1}\right)(t)\left(T^{-a+1}\right)\left(T^{-2}\right)e^{-\frac{t}{T}}dT \tag{8}$$

$$= t^a \int_0^\infty T^{-a-1}e^{-\frac{t}{T}}dT. \tag{9}$$

Now we set $a = d$ and isolate the polynomial in $t$, giving

$$t^{-d} = \Gamma(d)^{-1}\int_0^\infty T^{-d-1}e^{-\frac{t}{T}}dT. \tag{10}$$

This form closely resembles Equation 4, suggesting that $P(T) = \Gamma(d)^{-1}T^{-d-1}$. However, note that we did not rigorously define the domain of $t$ in the original formulation. In reality, we are only concerned with $t \geq 1$, since $0 \leq t < 1$ corresponds to distances of less than 1 word, and $t^{-d}$ explodes for values of $t$ near zero. We thus neither want nor need to approximate $t^{-d}$ for $0 \leq t < 1$.

We can adjust Equation 10, which is defined for $t \geq 0$, to reflect this limitation by simply substituting $t \to s + 1$, giving

$$(s+1)^{-d} = \Gamma(d)^{-1}\int_0^\infty T^{-d-1}e^{-\frac{s+1}{T}}dT \tag{11}$$

$$= \Gamma(d)^{-1}\int_0^\infty T^{-d-1}e^{-\frac{1}{T}}e^{-\frac{s}{T}}dT \tag{12}$$

$$= \int_0^\infty P(T)e^{-\frac{s}{T}}dT, \text{ where} \tag{13}$$

$$P(T) = \frac{T^{-d-1}}{\Gamma(d)}e^{-\frac{1}{T}} = \text{InverseGamma}(T; \alpha = d, \beta = 1). \tag{14}$$

## A.2  RELATIONSHIP BETWEEN FORGET GATE AND TIMESCALE

In Section 2.3, we showed that the forget gate bias controls the timescale of the unit, and derived a distribution of assigned timescales for the multi-timescale language model. After training this model,

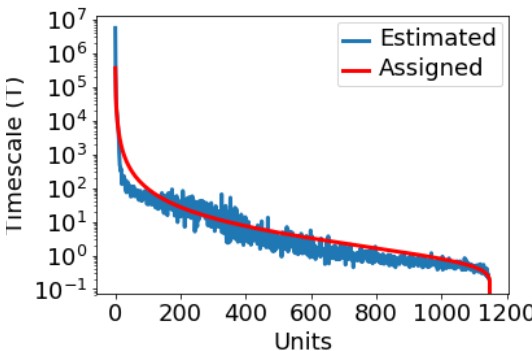

Figure 4: Estimated timescale is highly correlated with assigned timescale, shown for all 1150 units in LSTM layer 2 of the multi-timescale language model.

we tested whether this control was successful by estimating the empirical timescale of each unit based on their mean forget gate values using Equation 2. Figure 4 shows that the assigned and estimated timescales in layer 2 are strongly correlated. This demonstrates that the timescale of an LSTM unit can be effectively controlled by the forget gate bias.

Figure 1 shows estimated timescales in the LSTM language model from Merity et al. (2018) trained on Penn Treebank (Marcus et al., 1999; Mikolov et al., 2011). These timescales lie between 0 and 150 timesteps, with more than 90% of timescales being less than 10 timesteps, indicating that this network skews its forget gates to process shorter timescales during training. This resembles the findings by Khandelwal et al. (2018), which showed that the model's sensitivity is reduced for information farther than 20 timesteps in the past. Ideally, we would like to control the timescale of each unit to counter this training effect and select globally a distribution that follows from natural language data.

### A.3 FORGET GATE VISUALIZATION

To further examine representational timescales, we next visualized forget gate values of units from all three layers of both the multi-timescale and baseline language models as described in Section A.3. The goal is to compare the distribution of these forget gate values across the two language models, and to assess how these values change over time for a given input.

First, we sorted the LSTM units of each layer according to their mean forget gate values over a test sequence. For visualization purposes, we then downsampled these values by calculating the average forget gate value of every 10 consecutive sorted units for each timestep. Heat maps of these sorted and down-sampled forget gate values are shown in Figure 5. The horizontal axis shows timesteps (words) across a sample test sequence, and the vertical axis shows different units. Units with average forget gate values close to 1.0 (bottom) are retaining information across many timesteps, i.e. they are capturing long timescale information. Figure 5 shows that the baseline language model contains fewer long timescale units than the multi-timescale language model. They are also more evenly distributed across the layers than the multi-timescale language model. Figure 5b also shows the (approximate) assigned timescales for units in the multi-timescale language model. As expected, layer 1 contains short timescales and layer 2 contains a range of both short and long timescales. Layer 1 units with short (assigned) timescales have smaller forget gate values across different timesteps. In layer 2, we observe that units with large assigned timescale have higher mean forget gate values across different timesteps, for example the units with assigned timescale of 362 in 5b have forget gate values of almost 1.0 across all timesteps. Similar to the previous analysis, this demonstrates that our method is effective at controlling the timescale of each unit, and assigns a different distribution of timescales than the baseline model.

### A.4 WORD ABLATION

Another way to interpret timescale of information retained by the layers is to visualize the decay of information in the cell state over time. We estimated this decay with word ablation experiments as described in Section A.3.

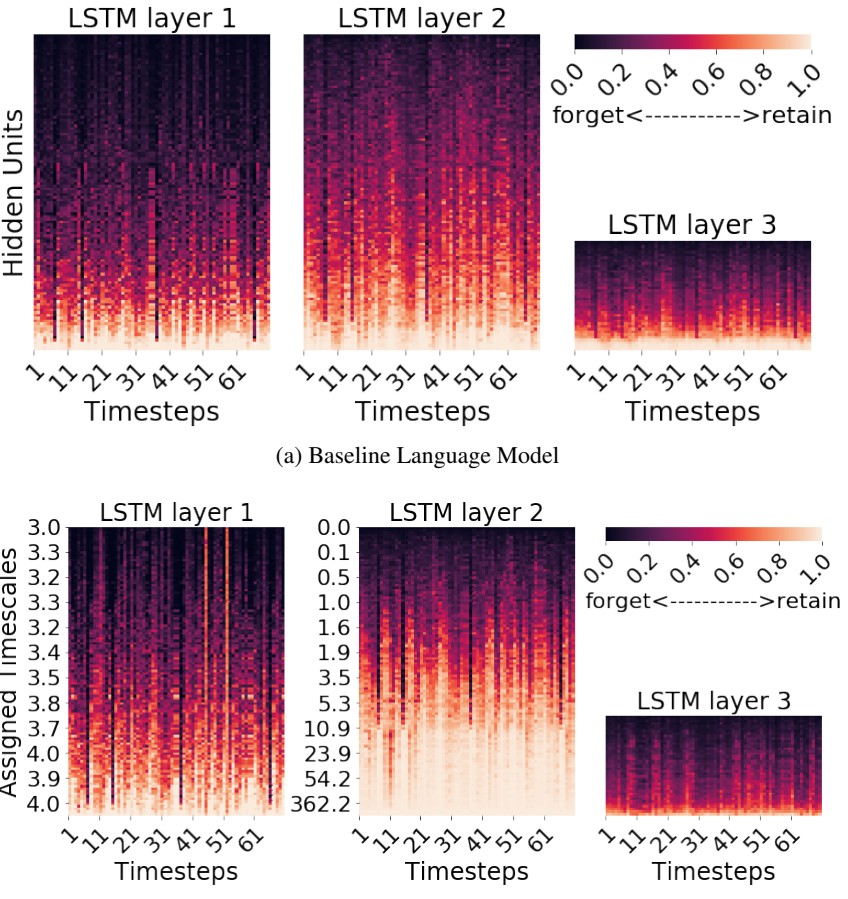

(a) Baseline Language Model

(b) Multi-timescale Language Model

Figure 5: Forget gate values of LSTM units for a test sentence from the PTB dataset. Units are sorted top to bottom by increasing mean forget gate value, and averaged in groups of 10 units to enable visualization. Figure 5b also shows average assigned timescale (rounded off) of the units.

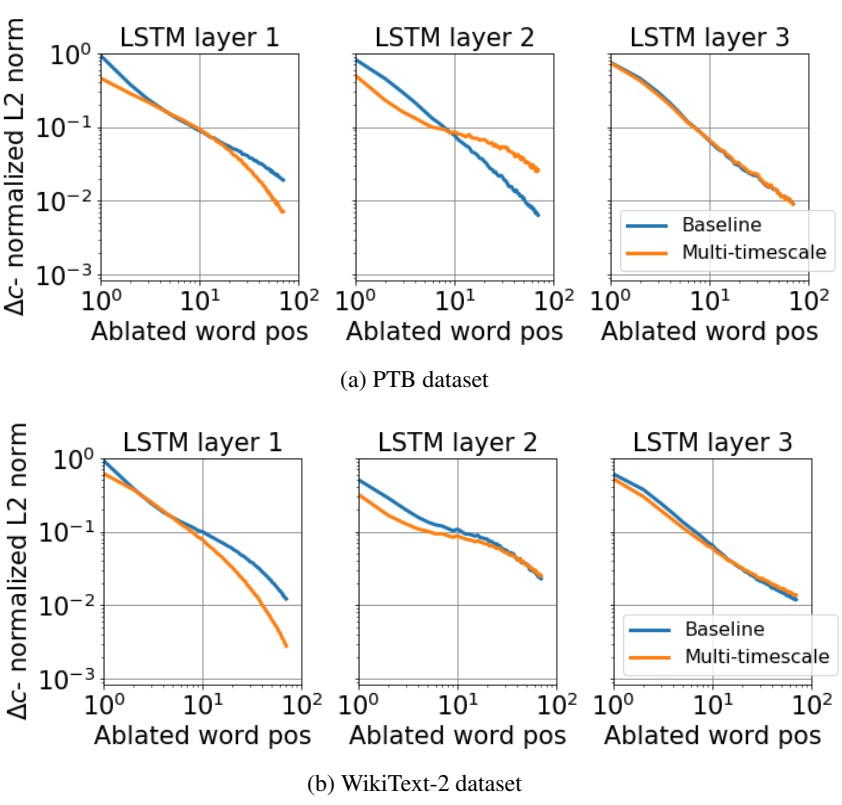

(a) PTB dataset

(b) WikiText-2 dataset

Figure 6: Change in cell state of all the three layers for both the baseline and Multi-timescale language models in word ablation experiment. A curve with a steep slope indicates that cell state difference decays quickly over time, suggesting that the LSTM layer retains information of shorter timescales.

In Figure 6, we show the normalized cell state difference averaged across test sentences for all three layers of both baseline (blue) and multi-timescale (orange) models. In the PTB dataset, information in layer 1 of the baseline model decays more slowly than in layer 2. In this case, layer 2 of the baseline model retains shorter timescale information than layer 1. In the WikiText-2 dataset, the difference between layer 1 and layer 2 of the baseline model is inverted, with layer 2 retaining longer timescale information. However, in the multi-timescale model the trend is nearly identical for both datasets, with information in layer 2 decaying more slowly than layer 1. This is expected for our multi-timescale model, which we designed to have short timescale dependencies in layer 1 and longer timescale dependencies in layer 2. Furthermore, the decay curves are very similar across datasets for the multi-timescale model, but not for the baseline model, demonstrating that controlling the timescales gives rise to predictable behavior across layers and datasets. Layer 3 has similar cell state decay rate across both models. In both models, layer 3 is initialized randomly, and we expect its behavior to be largely driven by the language modeling task.

Next, we explored the rate of cell state decay across different groups of units in layer 2 of the multi-timescale language model. We first sorted the layer 2 units according to their assigned timescale and then partitioned these units into groups of 100 before estimating the cell state decay curve for each group. As can be seen in Figures 7 and 8, units with a shorter average timescale have faster decay rates, whereas units with longer average timescale have slower information decay. While the previous section demonstrated that our manipulation could control the forget gate values, this result demonstrates that we can directly influence how information is retained in the LSTM cell states.

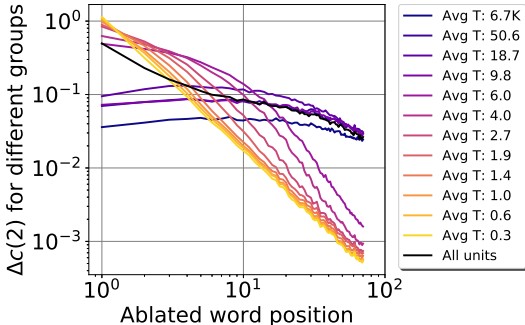

Figure 7: Change in cell states of 100-unit groups having different average timescale of layer 2 in Multi-timescale model in word ablation experiment for PTB dataset. As the assigned timescale to the group decreases the slope of the curve decreases indicating retained information of smaller timescale.

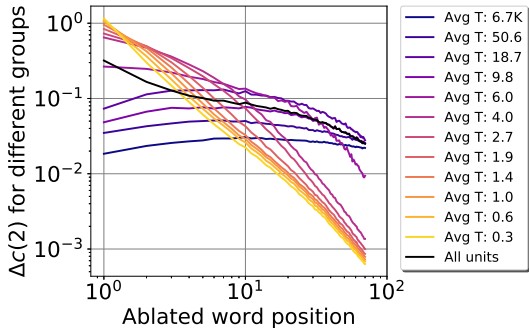

Figure 8: Change in cell states of 100-unit groups having different average timescale of layer 2 in Multi-timescale model in word ablation experiment for WikiText-2 dataset.

## A.5    SHAPE PARAMETER FOR INVERSE GAMMA DISTRIBUTION

We compared the performance of multi-timescale language model for different shape parameters in inverse gamma distribution. Figure 9 shows timescales assigned to LSTM units of layer 2

corresponding to different shape parameters. These shape parameters cover a wide range of possible timescale distribution to the units. Figure 10 shows that multi-timescale models performs best for $\alpha = 0.56$.

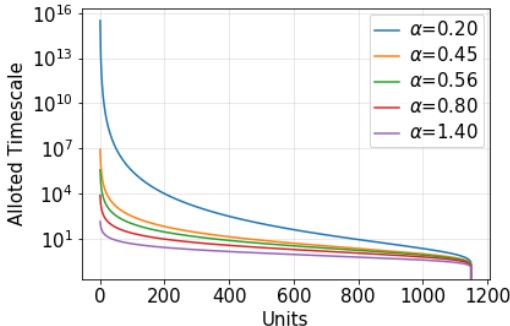

Figure 9: Assigned timescale to LSTM units of layer2 of multi-timescale language model corresponding to different shape parameter $\alpha$.

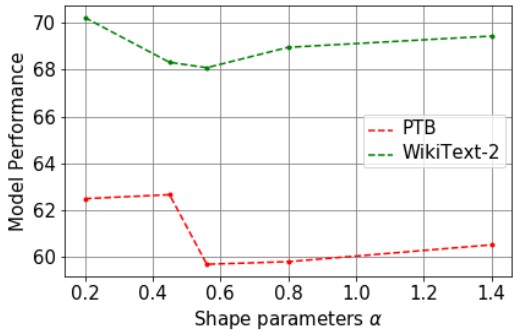

Figure 10: Performance of multi-timescale models for different shape parameters $\alpha$ on both PTB and WikiText-2 dataset.

### A.6 SELECTING THE TIMESCALES FOR EACH LAYER

With the purpose to select proper timescales to each layer in Section 3.1, we conducted experiments on designing LSTM language models with different combinations of timescales across the three layers. We found that layer 1 (the closest layer to input) always prefers smaller timescales within the range from 1 to 5. This is consistent with what has been observed in literature: the first layer focuses more on syntactic information present in short timescales Peters et al. (2018); Jain & Huth (2018). We also observed that the layer 3, i.e., the layer closest to the output, does not get affected by the assigned timescale. Since we have tied encoder-decoder settings while training, layer 3 seems to learn global word representation with a specific timescale of information control by the training task (language modeling). The middle LSTM layer (layer 2) was more flexible, which allowed us to select specific distributions of timescales. Therefore, we achieve the Multi-timescale Language Model in Section 3.1 by setting layer 1 biases to small timescales, layer 2 to satisfy the inverse gamma distribution and thus aim to achieve the power-law decay of the mutual information, and layer 3 with freedom to learn the timescales required for the current task.

### A.7 INFORMATION ROUTING EXPERIMENTS ON WIKITEXT-2 DATASET

We also performed the information routing experiments for multi-timescale model trained on WikiText-2 dataset. Figure 11 shows timescale-dependent routing in the model, same as what we observed for PTB dataset in Section 3.2.4.

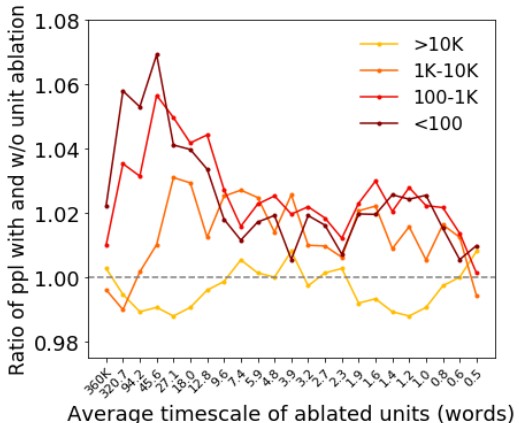

Figure 11: Information routing across different units of the multi-timescale LSTM for WikiText-2 dataset.

Table 2: Perplexity of the baseline and multi-timescale models over 5 different training instances. Values are the mean and standard error over the training instances.

| Dataset | Model | Performance |
|---------|-------|-------------|
| PTB | Baseline | $61.64 \pm 0.28$ |
| | Multi-timescale | $59.63 \pm 0.18$ |
| WikiText-2 | Baseline | $70.23 \pm 0.24$ |
| | Multi-timescale | $68.33 \pm 0.12$ |

## A.8 ROBUSTNESS OF MODEL PERFORMANCE

We quantified the variability in model performance due to stochastic differences in training with different random seeds. Table 2 shows the mean perplexity and standard error across 5 different training instances. The variance due to training is similar across the two models.

## A.9 MODEL COMPARISON REPRODUCING A PREVIOUS REPORT

Previously, the LSTM baseline we used in this work was reported to achieve a perplexity of 58.8 without additional fine-tuning (Merity et al., 2018). To more closely reproduce this value, we retrained the models using the legacy version of pytorch (0.4.1) used to train the LSTM in the original paper. We report both the overall perplexity and the performance in each word frequency bin (Table 3). While the overall perplexity still falls short of the original value, we speculate this could be due to the CUDA version, which was not reported. Still, we find that the multi-timescale model significantly improves performance for the the most infrequent words, mirroring the results reported in the main text.

We also performed the information routing experiment on the multi-timescale model trained with the older version of pytorch (Figure 12). Our results are very similar to those reported in the main text in Section 3.2.4, which demonstrate that timescale-dependent information is routed through different units.

Table 3: Perplexity of the baseline and multi-timescale models trained with a legacy version of pytorch. Performance is also reported separately for tokens across different frequency bins. Last row is the mean difference in perplexity (baseline − multi-timescale) across 10,000 bootstrapped samples, along with the 95% confidence interval (CI).

| **Dataset:** Penn TreeBank | | | | | |
| --- | --- | --- | --- | --- | --- |
| **Model** | **above 10K** | **1K-10K** | **100-1K** | **below 100** | **All tokens** |
| Baseline | 6.59 | 27.04 | 178.62 | 2069.81 | 58.98 |
| Multi-timescale | 6.79 | 26.68 | 167.97 | 1902.37 | 57.58 |
| Mean diff. | -0.2 | 0.37 | 10.65 | 167.27 | 1.4 |
| 95% CI | [-0.27,-0.14] | [-0.02,0.73] | **[7.77,13.78]** | **[120.22,215.97]** | **[0.91,1.86]** |

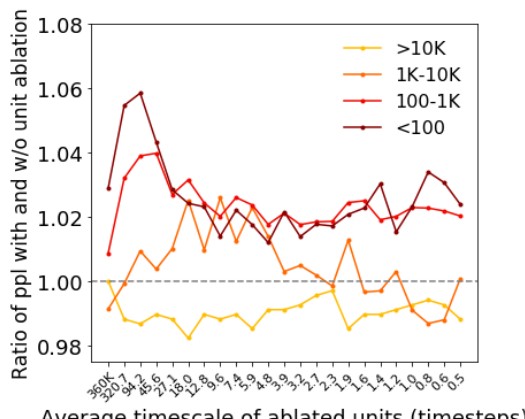

Figure 12: Information routing across different units of the multi-timescale LSTM trained with a legacy version of pytorch on the Penn Treebank dataset.

## A.10 THE DYCK-2 GRAMMAR

The Dyck-2 grammar (Suzgun et al., 2019) is given by a probabilistic context-free grammar (PCFG) as follows:

$$
S \rightarrow \begin{cases}
(S) & \text{with probability } p_1 \\
[S] & \text{with probability } p_2 \\
SS & \text{with probability } q \\
\varepsilon & \text{with probability } 1 - (p_1 + p_2 + q)
\end{cases}
$$

where $0 < p_1, p_2, q < 1$ and $p_1 + p_2 + q < 1$.

