# OpenReview forum: "Multi-timescale Representation Learning in LSTM Language Models"
_ICLR.cc/2021/Conference — ICLR 2021 Poster_

### Official Review · AnonReviewer1 · 2020-10-25
**Theoretically designed bias**

**Rating:** 7
**Confidence:** 4

**Review:**

With strong arguments I could be persuaded to change my score.

The paper investigates how well LSTM language models learn the known statistical temporal dependency distributions of languages, which follows a power law. The authors deduct the expected distribution of LSTM unit timescales and present a method for manually controlling them by setting the forget gate bias using an inverse Gamma distribution. They show that while the timescales of a standard LSTM LM follow the expected distribution, manually controlling it instead of learning it still increases LM performance, especially for infrequent words. The findings are also validated on a formal language, where the time scales are explicitly known.

What this paper excels at is a theoretical formulation of the proposed method and a motivation grounded in real contemporary problems in the field of modeling sequential data. However, I am not confident that the baselines and test suite employed by the authors is sufficient for me to accept their hypothesis.

The LSTM baselines in Table 1 are about 5 ppl points below the reportings of the paper they use as a baseline. Thus the authors "improvement" is actually significantly below the original paper from almost three years ago,which is a long time in this field, current models are now below half of the reported perplexity.

"Test data were divided into 100-word sequences and resampled with replacement 10,000 times. For each sample, we computed the difference in model perplexity (baseline − multi-timescale) and reported the 95% confidence intervals (CI) in Table 1. Differences are significant at p < 0.05 if the CI does not overlap with 0." - This test suite is unfamiliar to me. Please link a paper which uses the same test suite or follow the standard practices in the field of language modeling. Also if the authors are looking to argue for better handling longer time sequences, why cut samples off at 100 words? Shouldn't your model be better at longer sequences?

"This suggests that the performance advantage of the multi-timescale model is highest for infrequent words, which require very long timescale information" - It is not intuitive to me that rare words have more long-term dependencies than non-rare words? Also, I am unsure if the improvement in performance on rare words might just be attributed to the fact that the baseline model is significantly underperforming the original (Merity et al., 18) model.

Do you have confidence intervals for the DYCK language? Training models on DYCK-2 can be highly seed dependant. 5-10% improvement in performance seems like it could just be random noise.

I don't get Figure 3.

In general the paper is well-written and easy to understand.

UPDATE:

I do not believe that Merity 18's results are not reproducible. Just look at the vast amount of work after the publication that builds on top of the results. I don't doubt that your way of evaluating the performance of language models can lead to hypothesis testing, but it is not what the field employs thus your results are not comparable to others, which is why I cannot accept this paper. If your method really does work as well as you argue it should be no issue obtaining an improvement over the actual baseline.

UPDATE 2:

The new results with Merity's original benchmark leads me to increase my score from 4 to 7. I appreciate the effort in reproducing Merity's results.

---

> ### Author Response · Authors · 2020-11-18
> **Official response to Reviewer1**
>
> Thank you for taking the time to review our work. We would be happy to address your questions.
>
> Regarding baseline model performance: For the baseline LSTM language model, we used the publicly available code associated with the Merity et al. 2018 paper (https://github.com/salesforce/awd-lstm-lm). However, we used a later version of PyTorch (1.0.0 vs. 0.4), and so had to fix some minor issues (see comments in train_mts.py and weight_drop.py in the attached code, which has been updated further to work with PyTorch 1.7.0). These should not impact the results in any significant way. Despite using their code directly, we were unable to reproduce the perplexity numbers that they originally reported.
>
> Regarding the use of bootstraps to test significance: To test whether the multi-timescale model improved perplexity over the baseline model, we used a widely accepted statistical method, bootstrapping (Efron & Tibshirani, 1993). The idea is that a single perplexity value is only an estimate of the true value, which may vary depending on the data used to estimate it. Therefore, to estimate the range of the true perplexity, we can bootstrap over the data and generate confidence intervals by computing how perplexity changes over many random resamplings of the data. We first ran model inference over the entire test set, maintaining the hidden state between sequences, and measured the perplexity for every word. Since a model’s average perplexity is normalized by the number of words in a sequence, we divided the test data into sequences of 100 words, and resampled with replacement from these sequences. We then computed the model’s average perplexity by normalizing over the sequence length of 100 words and then averaging across sequences, following standard practice in language modeling. Note that since we ran stateful inference before dividing the test data, this procedure preserves the model’s ability to capture long-timescale information.
>
> Regarding long timescale information for rare words: We based our experiments on previously reported results that rare, infrequent words require longer timescales, or more prior context, to be successfully predicted by some model (Khandelwal et al., 2018; Griffiths et al., 2005; Rosenfeld, 1994; Iyer & Ostendorf, 1996). For example, Khandelwal et. al 2018 varied how much context was passed into an LSTM language model, and measured the decrease in the model’s ability to predict frequent (occurs >= 800 times in corpus) or infrequent (< 800 times) words. The model performs much worse on infrequent words given short contexts (5-20 words), requiring 150-250+ words to match performance on the frequent words.
>
> Regarding Figure 3: Following similar experimental logic as Khandelwal et al. 2018, we measured the model’s decrease in performance for very frequent words (occurring >10K times) up to very infrequent words (< 100 times). Instead of limiting the input to the model, we ablated groups of units based on their assigned timescale. This allowed us to test the hypothesis that long timescale units selectively process long timescale information, medium timescale units processes medium timescale, etc. The y-axis shows the decrease in model performance (higher ratio is worse PPL) as groups of units are ablated. The x-axis shows the average timescale of the ablated units (left is longer timescale). When ablating long timescale units, there is a selective decrease in performance for rare words (red lines, peak on left side of graph). Ablating medium timescale units selectively decreases performance for medium frequency words (orange line, middle peak). Ablating the shortest timescale units decreases performance for very frequent words (yellow line, far right point; also see far right points in Supp. Fig. 10). We have added some clarifications to this figure and its description in the document.
>
> Regarding seed-dependent training for the Dyck experiment: We modified the Dyck-2 figure to include the standard deviation over 20 repetitions training the models with different random seeds on the same dataset. The updated plot shows that the model improvement is not due to randomness in the model initialization or training.

---

> > ### Author Response · Authors · 2020-11-20
> > **Reproducing baseline model perplexity**
> >
> > We have retrained the models on PTB with the version of PyTorch that was used in Merity et al. 2018 (v0.4). We reproduced the original perplexity numbers (58.98 in our version compared to their 58.8, both without fine tuning). Still, our multi-timescale model is better overall (57.58).
> >
> > The small remaining discrepancy in the baseline model numbers could be due to the CUDA version (we used 10.1, theirs is unreported), as our other tests show that using a different CUDA version results in higher PPL, holding the other hyperparameters constant.
> >
> > Here we have recreated the top half of Table 1 (PTB results). While the multi-timescale model is no longer significantly better in the 1K-10K word frequency range, the improvement in perplexity for the two most infrequent categories is larger than in our original report. This shows that the multi-timescale model consistently improves language modeling performance for infrequent words.
> >
> > | Model           | above 10K     | 1K-10K       | 100-1K       | below 100       | All tokens  |
> > |-----------------|---------------|--------------|--------------|-----------------|-------------|
> > | Baseline        | 6.59          | 27.04        | 178.62       | 2069.81         | 58.98       |
> > | Multi-timescale | 6.79          | 26.68        | 167.97       | 1902.37         | 57.58       |
> > | Mean diff       | -0.2          | 0.37         | 10.65        | 167.27          | 1.4         |
> > | 95% CI          | [-0.27,-0.14] | [-0.02,0.73] | [7.77,13.78] | [120.22,215.97] | [0.91,1.86] |
> >
> > Looking forward to clearing any additional concerns you may have.

---

### Official Review · AnonReviewer2 · 2020-10-26
**Interesting approach for explicit control of LSTM unit timescale in natural language modeling**

**Rating:** 6
**Confidence:** 3

**Review:**

## Summary

This work investigates representational power of LSTM to model natural language, in particular how well it models temporal dependencies within text. They define a notion of timescale of each LSTM unit and analitycally show that LSTM memory exhibits exponential decay, while natural language tends (based on prior work) to decay following the power law. Based on this, they figure that LSTM memory may decay following the power law *if the timescales approximate samples from the particular Inverse Gamma Distribution*. To achieve that they propose the multi-timescale LSTM unit, where the desired timescale is explicitly controlled via the forget gate bias.

Authors empirically validate their theoretical claims and show improvements in language modeling (PTB, Wikitext2) over the baseline LSTM using the proposed multi-timscale LSTM. Importantly, they show how multi-timescale LSTM gives improvement in modeling rare words, which are known to require longer temporal dependencies.

## Strong points

1. This work investigates the important (though not that popular) question of discrepancy between the temporal dependencies existing in natural language and the abilities of models we use to learn these dependencies in practice.
2. The idea of including explicit control of the timescale (i.e. temporal horizon) of each LSTM unit is interesting and well-motivated.
3. Experiments use a formal language too in addition to natural language modeling which allows to check if proposed approach generalizes in case of exactly computable timescale distribution.

## Weak points

1. There is **no code** available. I was interested in the way how test set bootstraping was performed in the experiments (see comments below for details) and found out there is no code, which is really sad. I hope authors will submit the reproducible code in the near future.
2. Theoretical part gives some essential quantities while no derivations are given. Given that there is some free space left in the paper and the fact of unlimitied supplement material I don't see any reason to omit derivations (I struggle with some transitions between equations as you may found in the comments below).
3. I am not sure how useful is the proposed approach on new tasks given hardly tuned hyperparameters for the Inverse Gamma Distribution proposal and LSTM model architecture for each task (more details in the comments below).

## Recommendation

I vote for accepting **upon fixing major weak points**: uploading reproducible code with experiments and adding all derivations necessary for essential theoretical claims in this work. Overall this is decent work which will be useful for future research in studying representational power of models we are using to learn complicated dependencies of natural language.

## Questions

All the questions below are welcome to be used as **suggestions** to provide more details in the manuscript.

### Theoretical part

1. Eq.3: why can we simply average forget gates? Out of 'free input' regime $c_t$ from eq.1 would have more dependencies. Could you elaborate why can we estimate it like this.
2. How to solve Eq.4 as getting Inverse Gamma Distribution? I struggle to find an obvious/trivial solution, please elaborate this in the manuscript.

### Experiments
3. From 3.1.1. '*Training sequences were of length 70 with a probability of 0.95 and 35 with a probability of 0.05. During inference, all test sequences were length 70.*' Why are you making such explicit scheduling? Given that each training sequence from WT2 is some excerpt from wikipedia (often longer than 70 words), how do you deal with tails of sequences? More detailed description of data loading will be helpful.
4. Table 1: from my understanding columns with rare words attract most interest, and I wonder if you could add the varaiance among different training instances in Table 1 like you reported in the appendix (Table 2)? Or refactor Table 2 such that it has same freq based columns.
5. Did you think of tokenization other than word-level? BPE for example: it gives more balanced token distribution for WT2 due to absence of UNK token there. I wonder if improvements in terms of rare words PPL will hold. How do you think (speculatively)?
6. PTB results: >10K bin PPL got higher with your approach (also ratio drop below 1 in routing study). Why do you think this happens? Why do you think this does not happen with WT2 task? As I see with PTB the fixed forget bias underestimates true/gold high freq word probabilities on average, but why?

### Other

7. Is it possible to estimate/learn the IGD alpha parameter from the data itself of the task you work on? This grid search you provide makes me less convinced in how useful this approach is for the new task, where the $\alpha$ is not known.
8. How important is the model layers tuning you do? E.g. only specific layers have fixed forget bias, but others not, **why is that?** I am really interested in knowing if other ways of defining your model hurt the performance or keep it on the baseline level? I am sure this will be useful for all other readers too.
9. Is it possible to apply this timescale control for other units e.g. GRU (no explicit forget gate)?

---

> ### Author Response · Authors · 2020-11-18
> **Official response to Reviewer2**
>
> Thank you for this thoughtful review!
>
> Regarding the code: we have now included a folder containing our training and evaluation scripts for the multi-timescale LSTM. We have built our work on the code published by Merity et. al. 2018 (https://github.com/salesforce/awd-lstm-lm). We adapted the code to the current version of Pytorch (1.7.0), however we developed it on Pytorch 1.0.0. This includes code for all experiments except for the one involving the Dyck-2 language, which we are working on releasing.
>
> Regarding how bootstrapping was performed: Since the LSTMs are stateful, we had to ensure that even for bootstrapped sentences the context vector is accurate. To simulate this situation, we resampled predicted entropies for the test tokens. We first calculated the predicted entropy for each token in the test document and stored them in a vector (maintaining temporal order). For each bootstrapped test set, we randomly picked 100 consecutive entropies corresponding to the 100 consecutive tokens in the original test set resulting in a new test sentence. We kept doing this since the total token count in the bootstrapped test set was the same as our original test set. The implementation of the bootstrapping evaluation can be found in model_evaluation.py script under `bootstrap_evaluation()` function.
>
> We have answered your other questions in order here.
>
> 1. Of course it is true that, outside of the free regime, the LSTM cell state c_t can have highly complex dependencies and is not guaranteed to follow any simple rule. Still, averaging the forget gate values is useful because it tells us how long information is being retained in the cell state on average, across all time and all sequences tested.
>
> 2. We have included a detailed derivation in the updated paper (see Appendix A.1).
>
> 3. We followed Merity et al.’s training procedure exactly on these points. The sequence length is actually the length of the backpropagation through time (BPTT) window for gradients. The scheduling allows for variable BPTT for LM training which helps in efficiently using data. More details on the variable BPTT are in the Merity paper. Finally, the stateful nature of these LSTMs ensures that the model is still getting context for the sequences longer than BPTT length.
>
> 4. We agree that adding measures of variance across training instances would provide extra information about our model’s robustness. We have not yet been able to do this, but will add these results to Table 2 in the supplement by the end of the review period.
>
> 5. This is an excellent suggestion. We suspect that differences of PPL in different frequency bins would be diminished in this case (as the distribution of BP frequencies is much more compact than the distribution of word frequencies), although the overall PPL advantage for the multi-timescale model would likely endure. We have not yet tested this suggestion empirically, but will include as a future direction in the manuscript.
>
> 6. The effect described here is most pronounced in the ablation experiment, where we saw consistently decreased perplexity for high frequency words when ablating longer-timescale units. We suspect that this is due to the language model essentially defaulting to predicting high frequency words (or predicting according to unigram frequencies) in the absence of evidence for low frequency words.
> The resulting “reduction” in PPL is a somewhat counter-intuitive consequence of stratifying PPL by word frequency. For example, imagine a simple model that always assigned a probability of 99% to the word “and” and spread the remaining 1% probability mass over all other words. This model would have terribly perplexity overall, but would have very low perplexity if it was only evaluated at positions where the true word was “and”.
>
> 7. In our derivation, we found out that the inverse gamma parameter depends on the power law decay of the information in natural language. This decay curve can be estimated for any text corpus (or any other type of dataset) by computing mutual information between tokens (as in Lin & Tegmark, 2017). However, this computation is itself expensive and somewhat sensitive to hyperparameters. From an application point of view, performing a grid search over the IGD alpha parameter just like how we tune other hyperparameters is likely more suitable than extracting information decay behaviour from the dataset. However, it may also be possible to directly estimate alpha during model fitting via gradient backpropagation. This is something that we have not explored deeply yet, but will note as an interesting future direction.

---

> > ### Author Response · Authors · 2020-11-18
> > **Official response to Reviewer2 (cont'd)**
> >
> > (Cont'd from answers above)
> >
> > 8. Regarding model layers tuning (we also included this response to Reviewers #3 and #4): We selected the particular design of our network — with short timescales (3-4) in the first layer, Inverse Gamma-distributed timescales in the second layer, and unconstrained timescales in the third layer — based on a combination of examining the effect of word ablation (as in Figs. 7 & 11), reasoning about how timescales are combined across layers, and validation performance.
> >
> > First, we found from examining word ablation curves for different models that assigning timescales to the third layer had essentially no effect on the word ablation curve for that layer (similar to Figs. 7 & 11). We suspect that this is due to the proximity of this layer to the output and its smaller dimensionality, which constrain this layer much more effectively than layers 1 and 2. Thus for simplicity’s sake we elected to simply not constrain this layer.
> >
> > Second, we believe (although have not shown rigorously) that timescales are essentially additive across layers; i.e. if a unit with timescale 5 feeds into a unit with timescale 10, the effective timescale is 15. (However, note that in a real network with many units per layer, there are many possible combinations, so it is difficult to use additivity to reason about the timescales of actual units.) Thus, in order to obtain an Inverse Gamma distribution of overall timescales, we elected to only constrain one layer to follow that distribution, while keeping the other layers essentially flat with small timescale (layer 1) or unconstrained (layer 3).
> >
> > Third, we assumed that layer 1 would act as a “preprocessing” layer for the computations occurring in layers 2 and 3, so we constrained the timescales there to be short (3-4 words). We tested a number of different possibilities (T=2, 3, 4, 5, & mixtures) and selected the 50% mix of T=3 and T=4 units based on validation performance.
> >
> > 9. We believe the answer to this question is yes—in Tallec & Ollivier’s 2018 paper they argued that the same exponential timescale reasoning applies to both LSTM and GRU networks. It is relatively easy to see that the GRU would behave similarly to the LSTM in the free regime, albeit with forgetting controlled by the GRU “update gate” instead of the LSTM forget gate. Timescales in a GRU could thus be controlled by setting the bias of the update gate to the negation of the bias values we derived for the LSTM (Eq. 5).

---

> > > ### Comment · AnonReviewer2 · 2020-11-22
> > > **thanks for your detailed reply**
> > >
> > > I appreciate your detailed derivation in A1, it helped me to see the better connection. I have looked in the code and everything makes sense, the MTS module code is well commented and clear.
> > >
> > > All your replies support my initial vote to accept this submission!

---

### Official Review · AnonReviewer4 · 2020-10-28
**An interesting work that shed light on how to design a neural architecture inspired by a theory.**

**Rating:** 7
**Confidence:** 4

**Review:**

This paper points out the relationship between words in natural language usually follow the power law. Gated recurrent neural networks such as LSTMs excel in modelling natural language, however, the forgetting mechanism of LSTMs  is ruled by the exponential decay. This work demonstrates a way to engineer the forgetting mechanism of LSTMs to mimic the power law relationship that is more presented in natural language. By applying their technique, the modified LSTM model can do better in modelling rare tokens, which usually span for longer timescales, hence the model can score lower perplexities on less frequent words. The key contribution of the paper is the derivation which shows that the forget gates of LSTMs are subject to exponential decay in zero-input regime after the first input token is given. And the expected value of exponential decay functions exp(-t/T) can approximate the power law when T is sampled from the Inverse Gamma distribution.

The experiments demonstrate that that drawing T from the Inverse Gamma distribution is a natural fit for natural language. Then, the authors propose a multiscale LSTM model that exploits this property. Each timescale T is drawn from the inverse Gamma distribution, which essentially becomes a forget bias term and is fixed during the training. Multiple Ts are drawn to mimic the power law. The multiscale LSTM captures the right inductive bias to perform better in modelling less frequent words which might be useful to keep in the memory for longer time. The paper is well written and both the motivation and explanation of the approach are clear. The experiments are appropriately designed, and the results support the main claim nicely.

I do have some comments and questions on what's written in the paper though.
It is conventional to use notation h_{t-1} in the update rules for the input, forget and output gates since h_t is obtained from c_t, or at least clarify the update rule of the hidden state as h_{t+1} = o_t * tanh(c_t).

Instead of not learning the forget bias at all, have you tried adding a regularisation loss that forces the forget bias term to stay closer to the prior (initially drawn T from the Inverse Gamma distribution)?

What is the motivation of using smaller hidden size for the topmost layer for both the baseline and the multiscale LSTM?

For the multiscale LSTM, why were the timescales not sampled from the Inverse Gamma distribution and fixed during the training for the topmost layer? But instead, they were learned like the standard LSTM?

How were T=3 or T=4 chosen for the first layer of the multiscale LSTM? Were they found by evaluating performance on the validation set or were they chosen based on domain knowledge?

---

> ### Author Response · Authors · 2020-11-18
> **Official response to Reviewer4**
>
> Thanks for your time reviewing our work. We have modified the LSTM equations to follow the typical definitions using $h_{t-1}$.
>
> Regarding regularization loss instead of fixed bias: Applying a prior through a regularization term is a great suggestion that we have not yet tried, but will pose as a future direction.
>
> Regarding the smaller hidden size for the topmost layer: Our baseline model is the same as the Merity et al. 2018 work, which used 400 units in the last layer as well. The reason is that weights are shared between the embedding layer and the output layer after the last LSTM layer and before the softmax function.
>
> Regarding timescales in the 1st and 3rd LSTM layers (we also included this response to Reviewers #3 and #2): We selected the particular design of our network — with short timescales (3-4) in the first layer, Inverse Gamma-distributed timescales in the second layer, and unconstrained timescales in the third layer — based on a combination of examining the effect of word ablation (as in Figs. 7 & 11), reasoning about how timescales are combined across layers, and validation performance.
>
> First, we found from examining word ablation curves for different models that assigning timescales to the third layer had essentially no effect on the word ablation curve for that layer (similar to Figs. 7 & 11). We suspect that this is due to the proximity of this layer to the output and its smaller dimensionality, which constrain this layer much more effectively than layers 1 and 2. Thus for simplicity’s sake we elected to simply not constrain this layer.
>
> Second, we believe (although have not shown rigorously) that timescales are essentially additive across layers; i.e. if a unit with timescale 5 feeds into a unit with timescale 10, the effective timescale is 15. (However, note that in a real network with many units per layer, there are many possible combinations, so it is difficult to use additivity to reason about the timescales of actual units.) Thus, in order to obtain an Inverse Gamma distribution of overall timescales, we elected to only constrain one layer to follow that distribution, while keeping the other layers essentially flat with small timescale (layer 1) or unconstrained (layer 3).
>
> Third, we assumed that layer 1 would act as a “preprocessing” layer for the computations occurring in layers 2 and 3, so we constrained the timescales there to be short (3-4 words). We tested a number of different possibilities (T=2, 3, 4, 5, & mixtures) and selected the 50% mix of T=3 and T=4 units based on validation performance.

---

### Official Review · AnonReviewer3 · 2020-10-29
**Very interesting paper that combines scale-free property of natural languages with LSTM**

**Rating:** 8
**Confidence:** 4

**Review:**

This paper proposes a novel variant of LSTM by analyzing its behavior against
scale-free distributions generally found in natural languages. Since the
prediction of LSTM is essentially a convolution over each hidden unit, the
authors derived that the bias parameter should obey an inverse Gamma
distribution. This is a very neat and interesting result, which is also
validated by a number of experiments in natural languages with scale-free
distributions and artificially generated corpus with non scale-free
distributions.

My only question is the setup of the proposed LSTM: in Section 3.1.1, the
authors say that the first layer of LSTM has a fixed timescale, and only the
second layer has an inverse Gamma bias parameters. The third layer does not
have inverse-Gamma distribution and simply optimized.
Is this architecture necessary for the result? If so, why the third layer
should not have the proposed inverse-Gamma time scales?
Finally, in Figure 3, infrequent words actually use longer time scales, but
they also leverage short scales (i.e. red lines are U-shaped, not linear for
longer scales). I would like to know why this phenomenon happens.

That being said, this is a very interesting paper leveraging the structure of
LSTM and scale-free property of natural languages. In addition to Dyck
experiments, some other languages, such as a generation from PCFG or some
programming languages might be also interesting for experimentation.

---

> ### Author Response · Authors · 2020-11-18
> **Official response to Reviewer3**
>
> Thank you for your review!
>
> Regarding timescales in the 1st and 3rd LSTM layers (we also included this response to Reviewers #4 and #2): We selected the particular design of our network — with short timescales (3-4) in the first layer, Inverse Gamma-distributed timescales in the second layer, and unconstrained timescales in the third layer — based on a combination of examining the effect of word ablation (as in Figs. 7 & 11), reasoning about how timescales are combined across layers, and validation performance.
>
> First, we found from examining word ablation curves for different models that assigning timescales to the third layer had essentially no effect on the word ablation curve for that layer (similar to Figs. 7 & 11). We suspect that this is due to the proximity of this layer to the output and its smaller dimensionality, which impose more constraints on this layer than layers 1 and 2. Thus for simplicity’s sake we elected to simply not constrain this layer.
>
> Second, we believe (although have not shown rigorously) that timescales are essentially additive across layers; i.e. if a unit with timescale 5 feeds into a unit with timescale 10, the effective timescale is 15. (Note that in a real network with many units per layer, there are many possible combinations, making it difficult to use additivity to reason about the timescales of actual units.) Thus, in order to obtain an Inverse Gamma distribution of overall timescales, we elected to only constrain one layer to follow that distribution, while keeping the other layers essentially flat with small timescale (layer 1) or unconstrained (layer 3).
>
> Third, we assumed that layer 1 would act as a “preprocessing” layer for the computations occurring in layers 2 and 3, so we constrained the timescales there to be short (3-4 words). We tested a number of different possibilities (T=2, 3, 4, 5, & mixtures) and selected the 50% mix of T=3 and T=4 units based on validation performance.
>
> Regarding infrequent words and short timescales: We agree that this is an interesting result. Our speculation is that predicting infrequent words cannot rely solely on long timescale information, and requires some context at shorter timescales. The shorter timescale information might help predict its placement in a sentence with the correct part of speech, for example. This deserves further investigation in future work.
>
> Regarding other languages: The Dyck-2 grammar was generated given the definition in Appendix A.9, namely, from a probabilistic context-free grammar (PCFG). This is mentioned in Section 3.1.2.

---

### Decision · Program_Chairs · 2021-01-07
**Final Decision**

**Decision:**

Accept (Poster)

**Comment:**

The paper proposes matching the distribution of biases for an LSTM to estimates of long range mutual information from analyzing the statistics of languages. The authors shows empirical evidence that LSTMs seem indeed to be following such a distribution, using natural language and Dyck-2 grammar. They show that explicitly enforcing the distribution of biases in learning can actually help LSTM language models.
The reviewers had slight concern about some of the baseline numbers reported, but the authors took the time to address those concerns. Overall, it was an interesting and thought provoking paper that can provide a useful angle to consider when building recurrent models for a problem -- namely that of matching the properties / inductive bias of the model to that of the data.